# Large scale coherent magnetohydrodynamic oscillations in a sunspot

M. Stangalini [1✉], G. Verth[2], V. Fedun[3], A. A. Aldhafeeri [2,4], D. B. Jess [5,6], S. Jafarzadeh [7,8], P. H. Keys [5], B. Fleck [9], J. Terradas[10,11], M. Murabito [12], I. Ermolli[12], R. Soler[10,11], F. Giorgi[12] & C. D. MacBride [5]

Although theoretically predicted, the simultaneous excitation of several resonant modes in sunspots has not been observed. Like any harmonic oscillator, a solar magnetic flux tube can support a variety of resonances, which constitute the natural response of the system to external forcing. Apart from a few single low order eigenmodes in small scale magnetic structures, several simultaneous resonant modes were not found in extremely large sunspots. Here we report the detection of the largest-scale coherent oscillations observed in a sunspot, with a spectrum significantly different from the Sun's global acoustic oscillations, incorporating a superposition of many resonant wave modes. Magnetohydrodynamic numerical modeling agrees with the observations. Our findings not only demonstrate the possible excitation of coherent oscillations over spatial scales as large as 30–40 Mm in extreme magnetic flux regions in the solar atmosphere, but also paves the way for their diagnostic applications in other astrophysical contexts.

[1] ASI Italian Space Agency, Via del Politecnico snc, 00133 Rome, Italy. [2] Plasma Dynamics Group, School of Mathematics and Statistics, University of Sheffield, Sheffield, UK. [3] Plasma Dynamics Group, Department of Automatic Control and Systems Engineering, University of Sheffield, Sheffield, UK. [4] Mathematics and Statistics Department, Faculty of Science, King Faisal University, Al-Hassa, Hofuf 31982, Saudi Arabia. [5] Astrophysics Research Centre, School of Mathematics and Physics, Queen's University Belfast, Belfast, Northern Ireland, UK. [6] Department of Physics and Astronomy, California State University Northridge, Northridge, CA 91330, USA. [7] Max Planck Institute for Solar System Research, Justus-von-Liebig-Weg 3, 37077 Göttingen, Germany. [8] Rosseland Centre for Solar Physics, University of Oslo, Blindern, NO-0315 Oslo, Norway. [9] ESA Science and Operations Department, C/O NASA/GSFC Code 671, Greenbelt, MD 20771, USA. [10] Departament de Fisica, Universitat de les Illes Balears, E-07122 Palma de Mallorca, Spain. [11] Institut d'Aplicacions Computacionals de Codi Comunitari (IAC3), Universitat de les Illes Balears, E-07122 Palma de Mallorca, Spain. [12] INAF-OAR National Institute for Astrophysics, Via Frascati 33, 00078 Monte Porzio Catone (RM), Italy. ✉email: marco.stangalini@asi.it

The Sun, as well as many other stars, presents a series of oscillations which constitute the signature of global hydrodynamic resonant modes of the stellar structure[1] like pressure modes (p-modes) or gravity modes (g-modes). However, these are not the only resonances expected in the Sun. Almost 40 years ago, it was argued that the natural convective and dynamo processes at work beneath the solar surface provide the necessary forcing actions on the ambient solar photosphere to excite different magnetohydrodynamic (MHD) oscillations in magnetic fields pervading the solar atmosphere. This can lead to embedded MHD oscillations, which naturally represent the inherent resonant modes[2–6] of the flux tubes. MHD waves differ from purely acoustic waves since the restoring forces of magnetic tension and magnetic pressure also come into play. Since magnetic fields dynamically couple different layers of the solar atmosphere, MHD waves are thought to play a major role in heating the outer regions of the atmosphere to million-degree temperatures[7], and the acceleration of the solar wind[8–10].

To characterize MHD waves in magnetic flux tubes, the standard magnetic cylinder model has often been applied. The MHD modes of a magnetic cylinder are well understood and are identified by the number of wavelengths in the azimuthal and radial directions. Importantly, these modes can either be incompressible or compressible. For the incompressible modes, often called torsional/rotational Alfvén waves[11–13], the only restoring force is magnetic tension. For the compressible or magneto-acoustic modes, the restoring forces of magnetic tension, magnetic pressure, and gas pressure all play a part to varying degrees. The lowest order magneto-acoustic mode in the azimuthal direction is the so-called sausage mode. This has rotational symmetry about the flux tube axis and is propelled by both magnetic field and plasma compressions and rarefactions analogous to a peristaltic (or extensional) wave. The lowest order magneto-acoustic mode with azimuthal asymmetry is the kink mode and causes a bulk transverse displacement of the flux tube analogous to a flexural wave.

However, as of now, apart from a few single low order eigenmodes (i.e. sausage and kink), which were identified in small scale magnetic structures (diameters up to a few Mm)[14–17], several simultaneous resonant modes were not found at the same time in extremely large sunspots with diameters of tens of Mm. From an observational point of view, sunspots are found to be dominated by 5-m oscillation periods in the photosphere[18], which arise as a result of p-mode conversion by the magnetic field[19,20]. These 5-m oscillations dominate in these magnetic structures, hampering the identification of other periodicities (if they exist).

In this work we analyze a 40 Mm sized sunspot observed on May 20, 2016 and report the largest coherent oscillations observed in a sunspot, which agree with numerical modeling. The results can be interpreted as a superposition of several resonant modes of the magnetic flux tube.

## Results

An exceptionally intense sunspot in active region AR12546 (Fig. 1) was observed for more than three hours continuously by the high resolution Interferometric Bidimensional Spectropolarimeter (IBIS[21]) at the DST (Dunn Solar Telescope; New Mexico, USA). This is one of the largest sunspots of the last 20 years. In Fig. 1, we show a contextual 1600 Å image captured by the Atmospheric Imaging Assembly (AIA) onboard the NASA Solar Dynamics Observatory (SDO; panel a), acquired at the same time of the IBIS observations (panel b), giving the impression of the extreme size of this sunspot. Notably, the magnetic field strength at the center of the umbra exceeds 3.5 kG, which is a rare occurrence[22,23]. These observations represent ideal unique conditions for the investigation

of MHD modes. Their uniqueness resides in high-spatial and temporal resolution of the data and the intense magnetic field strengths which, by significantly reducing the 5-m (≈3 mHz) power associated to the conversion of the global acoustic resonances of the stellar structure (i.e. p-modes[19]), allows the investigation of the internal wave dynamics of the sunspot.

**Analysis of the oscillations in the umbra.** The $B - \omega$ diagram of Fig. 2a shows the modification of the power spectrum of the line-of-sight (LOS) Doppler velocity, as one approaches the sunspot's umbra. Here we clearly see the transition from p-modes dominating the photosphere outside the sunspot, to the onset of a series of peaks immediately inside the umbra-penumbra (u-p) boundary[24] (see also panel b), whose non-uniform scaling may be due the combination of both atmospheric stratification within the umbra and the nature of the sub-surface driver itself[25].

Filtering of the LOS velocity in both spatial and temporal domains at the most significant frequencies as in[26] (see Fig. 3), allows the contribution of global resonant modes to be isolated from localized fluctuations detected in the same magnetic structure[27–29]. The oscillations associated with the global eigenmodes account for up to 10% of the total variance of all the Doppler fluctuations observed in the umbra. The filtering reveals a spatially coherent oscillation (Fig. 1 panels c and e), with concentric annulii in antiphase (see also Supplementary Movie 1).

**Modal decomposition of the oscillations.** To understand these particular oscillation patterns, we modeled the observed umbra as an MHD waveguide to determine its eigenmodes. The numerical solution to the eigenvalue problem results in the orthogonal modal basis, shown in Fig. 4, which is strongly determined using the observed cross-section shape of the umbra.

The modal projection of the LOS velocity on the modal basis (Fig. 5) shows that most of the energy is contained in the first 10–15 eigenmodes, but at least 30 modes are needed to fully reproduce the observed signal before reaching a plateau in the reconstruction error. For comparison with numerically modeled superposition of the most powerful modes, in Fig. 1 (panels c-e) we show the instantaneous observed oscillatory pattern in the umbra alongside the MHD slow body mode model that uses the superposition of first 9 most energetic orthogonal eigenfunctions (panel d and f). The observed velocity pattern can be then well approximated by a linear combination of the most energetic modes (see Fig. 5). In panel (a) of Fig. 5, we also plot the reconstruction error as a function of the number of modes used. In this plot we show both the reconstruction accuracy with the ordering of the modal basis as in Fig. 4 and with the eigenmodes ordered by energy (i.e., fraction of variance of the signal). Here we see that a plateau is reached after only using 30–35 modes and no benefit in the reconstruction is obtained by adding more modes, meaning that these latter (i.e., 10–15 residual modes of the basis) are not able to improve the representation of the signal. From this plot it is clear that several modes are excited in the magnetic structure, and they coexist at the same time.

In relation to the magnetic cylinder model, which has been widely used for the interpretation of MHD modes in the Sun's atmosphere, it is found that the most dominant modes are sausage-like, although the irregular cross-sectional shape of the umbra physically prevents true azimuthal symmetry in their spatial pattern. Both the fundamental and first radial overtone of this mode have the largest contribution. This demonstrates that several eigenmodes have been simultaneously excited in the umbra and that the shape of the u-p boundary dominates their spatial structure.

The temporal analysis of the coefficients of the modal projection (Fig. 6) reveals that, in contrast to p-modes, the

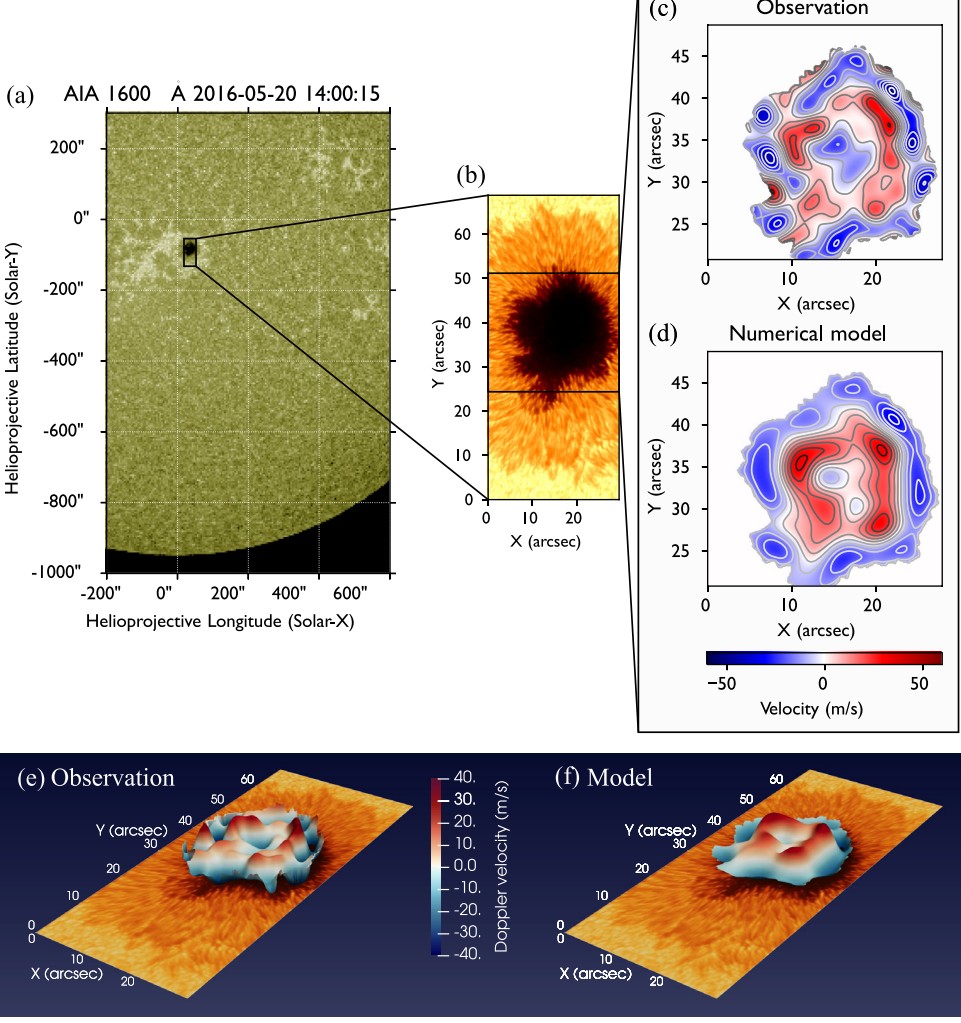

**Fig. 1 Detection of high radial order MHD oscillations in a sunspot. a** AIA/SDO full disk image captured in the 1600 Å band, showing the IBIS FOV as a solid black rectangle. **b** High resolution intensity image acquired by IBIS in the continuum in proximity of the Fe I 6173 Å spectral line. **c–f** Example of an instantaneous map of the filtered Doppler velocity, derived from the IBIS Fe I 6173 Å spectral imaging sequence. In the sunspot umbra concentric oscillatory annulii are visible. These rings oscillate in antiphase (see Fig. 6 and the Supplementary Movie 1 to see the evolution of the eigenmodes). **d–f** Numerically modeled LOS velocities obtained by the superposition of 9 orthogonal eigenmodes assuming that the umbra-penumbra (u-p) boundary is fixed. Panels **e**, **f** show the same field-of-view of panel **b** and are combining the information presented in panels **b–d**, where velocities (panels **c**, **d**) are overlaid on the observation (panel **b**). The pattern is extremely sensitive to the exact shape of the umbra, and this needs to be taken into account when generating the numerical model (see methods subsection Numerical modeling and modal reconstruction). Due to this, the oscillatory rings are distorted by the shape of the umbra, departing from the perfectly circular shape found in the case of the standard magnetic cylinder model. The numerical model incorporates the effect of superposition of several eigenmodes, which are simultaneously excited in the umbra. In relation to the standard magnetic cylinder model, the dominant modes are found to be sausage-like and contain both the fundamental and the first radial overtone. Panels **e**, **f** show the same field-of-view of panel **b**.

eigenmodes of the umbra are highly non-stationary and they all have a similar power spectrum. However, despite this, some of the modes are more persistent than others, with amplitudes above the noise threshold for up to 60% of the total duration of the data set.

The temporal evolution of the projection coefficients of the modal reconstruction shows a non-stationary behavior of all the energetically dominant modes (panel a in Fig. 6). In the same figure (panel b) we also show the persistence of each mode, which is defined as the time where the amplitude of the coefficients is above the noise level. In this plot we observe that the modes are non-stationary and that their persistence can be up to almost 60% of the total duration of the observations.

Finally, in panel (c) we show the power spectrum of each modal coefficient of the modal reconstruction. Here we see that they all share a similar spectrum.

**Dispersion relation of the modes**. The B-ω diagram of Fig. 2 presents several peaks with less power in addition to the most prominent ones, spread over the range of frequencies explored. The detectability of the spectral features depends on a number of factors, including the frequency resolution imposed by the length of the time series, the time cadence and the signal-to-noise ratio of the individual frequencies, however, the wide range of detected frequencies associated with each mode is also evident in Fig. 6 from the spectral analysis of the projection coefficients of the modal basis. This fact can be better seen in Fig. 7, where we compare the theoretical dispersion relation, obtained from the MHD model, with the one obtained from the observations (see methods subsection Dispersion relations for more details). Given the uncertainties, these two independent approaches to reproducing the dispersion relation for the

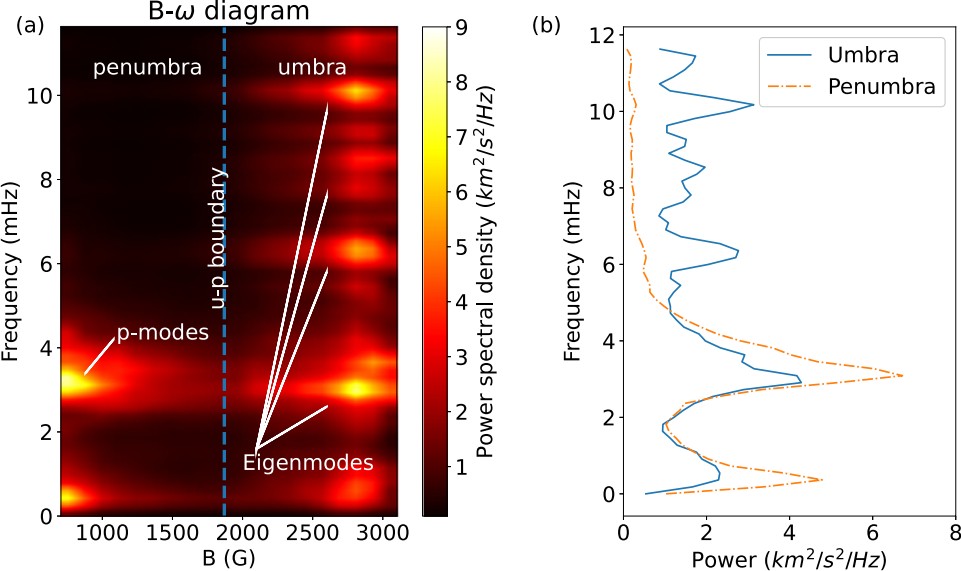

**Fig. 2 Transition from p-modes to resonant modes. a** $B - \omega$ diagram of the Fe 6173 Å Doppler velocity fluctuations. The vertical blue dashed line indicates the expected position of the u-p boundary (B=1867 G,[24]). The penumbra is dominated by p-modes at ≈ 3 mHz which, as expected, are converted as one approaches the inner part of the sunspot (increasing magnetic field strength), with a consequent reduction of power in the same band. Inside the umbra, several eigenmodes are visible. Here we consider the LoS component of the magnetic field. In addition to the most prominent peaks, around 2.5, 5.5, and 9 mHz, a number of additional spectral features is also visible, although with a smaller amplitude. **b** Average power spectrum of the penumbra (orange dash-dotted line) and the umbra (continuous blue line). Source data are provided as a Source Data file.

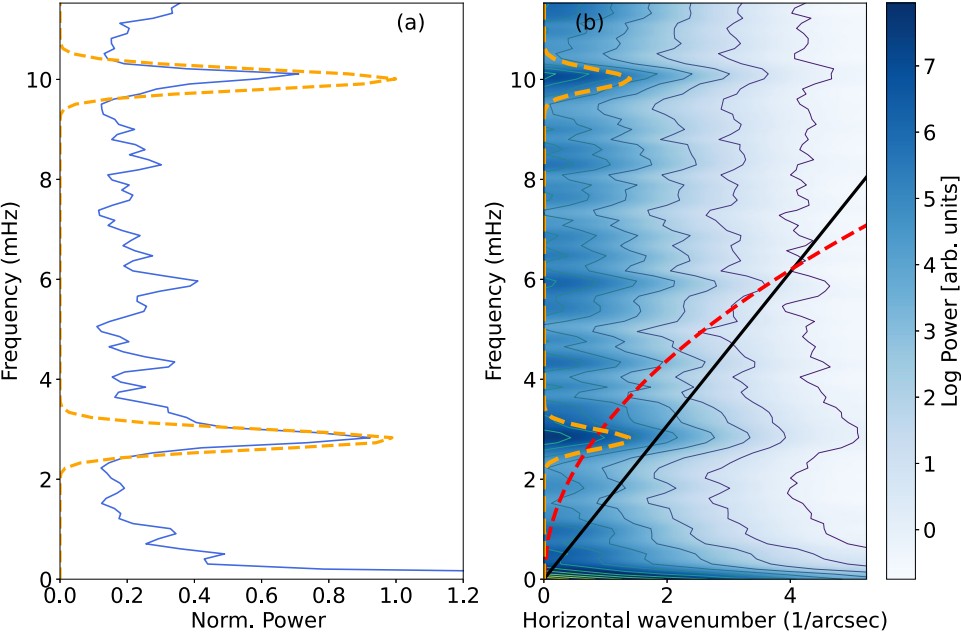

**Fig. 3 k– ω diagram. a** Global power spectrum of the Fe 6173 Å Doppler velocity fluctuations in the umbra of the sunspot. Two distinct peaks at 2.4 mHz and 9 mHz display confidence levels larger than 95%. **b** k-ω diagram of the Doppler velocity perturbations in the umbra of the sunspot revealing how the power of the oscillations is distributed in both the temporal and spatial frequency domains. Here we see that, in correspondence to the two frequencies identified, there exists an enhancement of power which results in horizontal bands. The red dotted lines indicate the 95% confidence level, obtained from a Monte Carlo randomization test, in which the data cube represented by the sequence of Doppler velocity maps are repeatedly shuffled, one thousand times, in order to break spatial and temporal correlations and assess the probability of noise reproducing the features observed in the diagram. The theoretical acoustic fundamental mode (f-mode) is represented by the red-dashed line and depends on both the gravitational acceleration g and the horizontal wavenumber k; $\omega = \sqrt{gk}$. The black line represents the expected propagation at the sound speed ($\omega = c_s k$, considering a typical sound speed of $c_s = 7$ km/s). These two lines are overplotted for reference. In both panels, the orange dashed lines represent the shape of the filter used to visualize the sunspot mode in Fig. 1 (panels c and d). Source data are provided as a Source Data file.

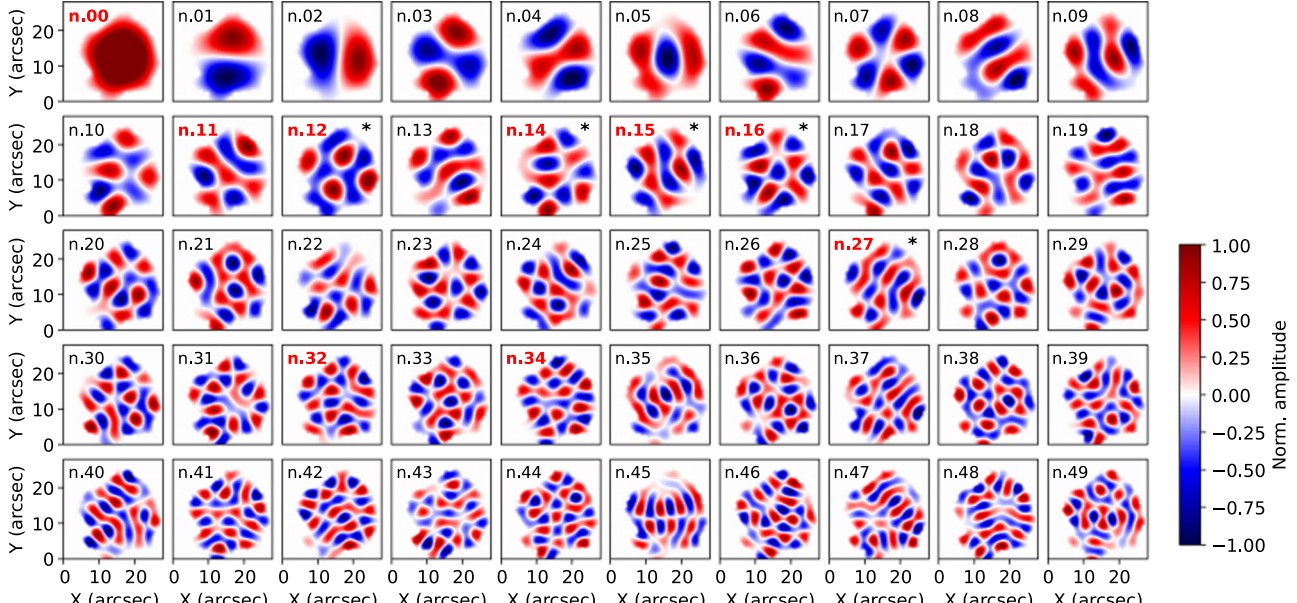

**Fig. 4 Computed eigenmodes of the sunspot.** Modal basis used to decompose the umbral velocity pattern. Red labels highlight the representative modes used to reconstruct the observed pattern in Fig. 1 (panels c and d), while the black asterisks represent the most energetic modes over the entire observation. The modal basis is obtained by solving the equation of the $v_z$ velocity perturbation considering the cross-sectional shape of the umbra. Modal reconstruction of the observed velocity pattern is obtained by decomposing it as a linear combination of the computed modes. See methods subsection Numerical modeling and modal reconstruction for details.

observed MHD slow body wave modes are in reasonable agreement.

## Discussion

The $B - \omega$ diagram of Fig. 2 captures in one single plot the essence of years of theoretical predictions, showing the transition from global acoustic p-modes to a spectrum of localized MHD eigenmodes, shown by a series of spectral feature which appear immediately inside the umbral boundary, revealing the inherent resonances of its magnetic structure.

This result demonstrates that sunspots are not always dominated by a 5-m signal, which is a natural consequence of the conversion of p-modes, but possess their own natural eigenmodes that, although they have a smaller amplitude, they can still be identified by means of filtering techniques. The signal associated to these oscillations are indeed identified thanks to a combination of Doppler and spectropolarimetric information which are simultaneously exploited to reveal the variation of the dynamics as a function of the magnetic flux.

It is interesting to note that a propagating surface mode localized at the u-p boundary was also identified[30] in the same sunspot with a frequency around 2.5 mHz, which is consistent a dominant power band in the $B - \omega$ diagram of Fig. 2.

The agreement between the observed velocity pattern and the MHD model enables the interpretation of the high radial order spatially coherent oscillations in terms of superposition of several eigenmodes, and identifies a more accurate approach for the prediction of resonant modes in complex solar magnetic structures, such as sunspots and pores, taking into account the actual cross-sectional shape in the calculation of their eigenfunctions. In this regard, it is worth noting how such a precise match is obtained by the superposition of only a limited number of eigenmodes, suggesting that the MHD modeling can realistically recreate the physical traits of the resonant oscillations of a sunspot umbra.

In the broader solar context, the success of global helioseismic inversions to determine the Sun's internal structure has depended on the large number of p and f modes (global acoustic and surface gravity modes) detected, up to degrees of several thousand[31].

Similarly to what is done with global resonances of the stellar structure, which are nowadays commonly employed to study the interior of the Sun[31,32] and other stars[33–35], the study of the global resonances of magnetic structures such as sunspots or star spots can indeed provide crucial information about a magnetically dominated stellar atmosphere.

Coronal seismology has been already proposed[36] and applied as a diagnostic tool for determining the physical conditions of the coronal environment[25,37–39]. However, nothing similar was possible in the lower solar atmosphere where, as our results suggest, resonant oscillations are hidden by large amplitude incoherent fluctuations due to, e.g., local temperature fluctuations or p-modes leakage. Hence the present analysis, not only provides evidence of the co-existence of many MHD modes and their radial overtones in a sunspot umbra, but also paves the way to use this methodology to advance the field of local magneto-seismology. Our results not only provide constraints for the study of the sub-surface structure, but are also key for the interpretation of the complex power spectra associated with sunspot oscillations and pores in the lower solar atmosphere.

Finally, it is worth noting how such large coherent oscillations can be excited, even in extremely large magnetic structures like the one observed, and result in such a clear LOS velocity Doppler signal in the photosphere. It is a well-known fact that the stellar surface activity has a direct impact on exoplanet detection and their characterization[40–43]. Our results suggest that, in addition to surface convection, MHD oscillations should also be also considered in this regard, as well as in stellar seismology where they could represent a substantial source of noise, especially for stars with extremely large stellar spots.

## Methods

**Data acquisition and calibration.** The data set used in this work was acquired starting from 13:39 UTC on May 20[th], 2016 with the Interferometric BIdimensional Spectrometer (IBIS) at the Dunn Solar Telescope (DST). The complete

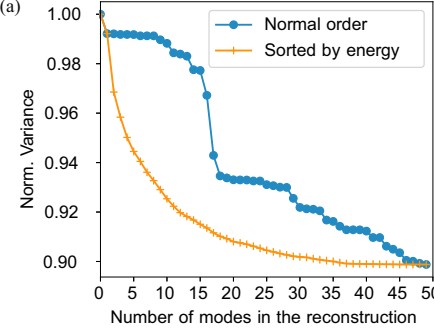

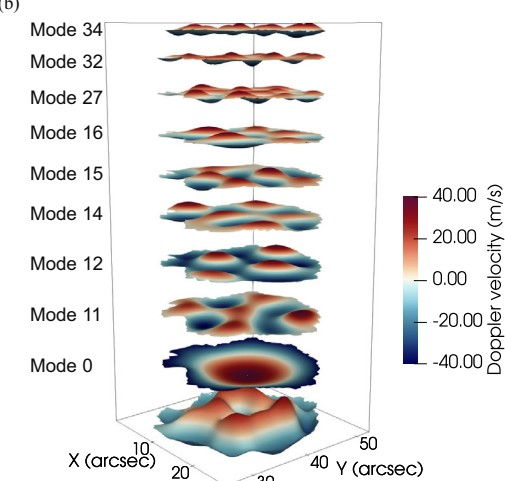

**Fig. 5 Modal decomposition. a** Reconstruction error as a function of the number of eigenmodes with normal order as in Fig. 4 (blue), and by energy (orange). **b** Example of modal reconstruction with the 9 modes of Fig. 4. The final reconstructed model, first layer at the bottom, is the weighted superposition of these eigenmodes which are represented with a vertical offset for better visualization. Source data are provided as a Source Data file.

sequence of data can be downloaded from the IBIS Archive (through a query including the observing date e.g., 19–21 May 2016). The total length of the observation is 184 min and consists of a sequence of full-Stokes spectropolarimetric scans of the Fe I 6173 Å spectral line of AR12546. Each scan consists of 21 spectral points acquired with a spectral sampling of 20 mÅ. The integration time of each spectral image was 80 ms.

The cadence of the reduced data is 48 s. AR12546 was close to the disk center [7° S, 2° W] and seeing conditions were stable. During the observation the adaptive optics (AO) system was used to mitigate the effects of atmospheric turbulence.

The theoretical diffraction-limited spatial resolution is 0.16 arcsec at the wavelengths considered here. Calibration procedure consist of standard flat fielding, dark subtraction, polarimetric calibrations, and finally image restoration with the MOMFBD code[44] to reduce the impact of residual seeing aberrations.

*B-ω analysis.* The B-ω diagram of Fig. 2 is constructed by averaging the power spectrum of the LOS velocity oscillations in pixels with almost the same magnetic flux (100 G bins). This is done from the minimum magnetic flux measured in the FOV (i.e., pixels outside the sunspots) to the maximum of B reached in the umbra. This is done to study the variation of the power spectrum as a function of the magnetic field regime,to explore the wave dynamics from the outside to inside the sunspot. The average magnetic flux is obtained from spectropolarimetric inversions of all the spectral scans composing the data sequence and performed with the VFISV code[44].

The power spectrum in each pixel is computed by means of the Welch method[45]. In this case, the data sequences are subdivided into segments of approximatively 75 min (i.e. 96 spectral scans). The power spectrum is then obtained by averaging the power spectra of the single segments composing the total length of the LOS velocity time series in each pixel. Although reducing the frequency resolution, dividing the data series into segments allows an improvement of the signal-to-noise ratio in the frequency range above ≈1.5–2 mHz.

*k-ω filtering.* As in Jess et al.[26], the LOS Doppler velocity is filtered in the k-ω space. This is done by a fast Fourier transform (FFT) of the data cube after applying a Hamming apodization window to limit edge effects in the Fourier space. The Fourier spectrum is then averaged in $k_x$ and $k_y$ in 10 pixels width concentric annuli. In Fig. 3 we show the resulting k-ω diagram of the entire data sequence of the photospheric Doppler velocity and the filter profiles used (dashed line, panel b) in the Fourier space. Differently from B-ω diagram, the plot visualizes the distribution of the power in k and ω. In agreement with the findings of Jess et al.[26] in the case of low radial order modes, the diagram displays different horizontal bands. A Monte Carlo randomization test employing 1000 runs has shown that the peaks at 2.7 mHz and 9 mHz in the k-ω diagram have a confidence level higher than 95%, thus they are selected for the filtering. The result of this filtering is shown in Fig. 1, panels (c) and (e) and Supplementary Movie 1.

*Numerical modeling and modal reconstruction.* Since there is no preferred geometry for the irregular cross-sectional shape of the umbra and a numerical approach is required to find the eigenfunctions we choose to work in Cartesian coordinates. Here we assume the photosphere as the xy-plane, and the vertical axis oriented along the vertical.

The spatial structures of the eigenfunctions are mainly physically constrained by the cross-sectional shape of the sunspot. For this reason, we solve the equation of the $v_z$ velocity taking into account the shape of the umbra and, consistently with the observations, setting $v_z = 0$ at the u-p boundary. Here we are assuming uniformity of the sound and Alfvén speeds within the umbra, something which is supported by the complete absence of umbral dots or other possible plasma inhomogeneities. The shape of the umbra is estimated by considering the observed $v_z = 0$ boundary of the filtered Doppler velocity field, thus represents a dynamical estimate of the sunspot's shape based on the eigenmodes hosted within the umbra. This is also very close to the definition of u-p boundary as in[24]. Assuming only linear MHD perturbations of a magnetic flux tube, the equation governing the $v_z$ velocity perturbations is the Helmholtz equation, i.e.,

$$\frac{\partial^2 v_z}{\partial x^2} + \frac{\partial^2 v_z}{\partial y^2} - m_0^2 v_z = 0, \qquad (1)$$

here $m_0^2 = \frac{(k_z^2 c_s^2 - \omega^2)(k_z^2 v_A^2 - \omega^2)}{(c_s^2 + v_A^2)(k_z^2 c_T^2 - \omega^2)}$, $k_z$ is the vertical wavenumber, $\omega$ is the angular (or cyclic) frequency, $c_s$ is the sound speed, $v_A$ is the Alfvén speed and $c_T = \frac{c_s v_A}{\sqrt{c_s^2 + v_A^2}}$ (eq. 3) is the tube (or cusp) speed. The observed oscillations are consistent with a body mode, hence $m_0^2 < 0$ (a surface mode has $m_0^2 > 0$ and would be evanescent in the umbra, while at the u-p boundary it would and reach its maximum amplitude). The eigenvalue $m_0^2$ depends on implicit parameters, specifically $k_z$, $c_s$ and $v_A$. However, in order to accurately model the spatial structure of the velocity pattern one can calculate the numerical values of $m_0^2$, and solve the eigenvalue problem with the above boundary conditions. The numerical solution to the eigenvalue problem results in the orthogonal modal basis shown in Fig. 4 which strongly depends on the actual shape of the sunspot.

In order to solve numerically the eigenvalue problem, the standard *delsq* and *eigs* routines on MATLAB were used. The *delsq* routine discretizes the Laplacian operator in the governing Helmholtz equation and for numerical stability a 5-point finite difference was chosen. To model the eigenfunctions with sufficient resolution 1043 discrete points were chosen around the u-p boundary and a 131240 ×131240 matrix was generated by *delsq* to use as an input for *eigs*, which is an implementation of the ARnoldi PACKage (ARPACK,[46]) in the MATLAB environment, to find the first 50 eigenvalues and eigenfunctions shown in Fig. 4.

The computed eigenmodes are used as the modal basis to decompose the observed 2D LOS velocity signal This approach is similar to other filtering techniques, like for instance proper orthogonal decomposition[47], except that here the modal basis is constructed upon physical assumptions (i.e., the modal basis are the actual eigenmodes of a magnetic flux tube with the same cross-sectional shape as the observed umbra), thus offering a direct interpretation of the wave modes.

In order to reconstruct the observed instantaneous velocity field, see for instance Fig. 1 panels (d) and (f), we project the pattern of the Doppler velocity onto the numerical modal basis containing the eigenfunctions of oscillation of the magnetic structure. The observed velocity pattern can be well approximated by a linear combination of the most energetic modes. Our method can be seen as a modal reconstruction technique in which, differently from other statistical approaches, e.g., proper orthogonal decomposition[47], the modal basis is computed from a physical model, thus each mode has a unique physical interpretation.

**Dispersion relations**. By means of spectropolarimetric inversions of the Fe I 6173 Å data by using the NICOLE code[48] we estimated the density, gas pressure and magnetic flux in each pixel within the umbra and estimated the main characteristic speeds; namely the Alfvén, sound and hence the tube speed in the same region. Their probability density function (PDF) is shown in Fig. 8. These characteristic speed values are then used in the eigenvalue $m_0^2$ calculated for each mode to find the dispersion relation, shown for the first 6 modes in Fig. 7, where it is possible to see that frequencies can be excited right across the spectral range

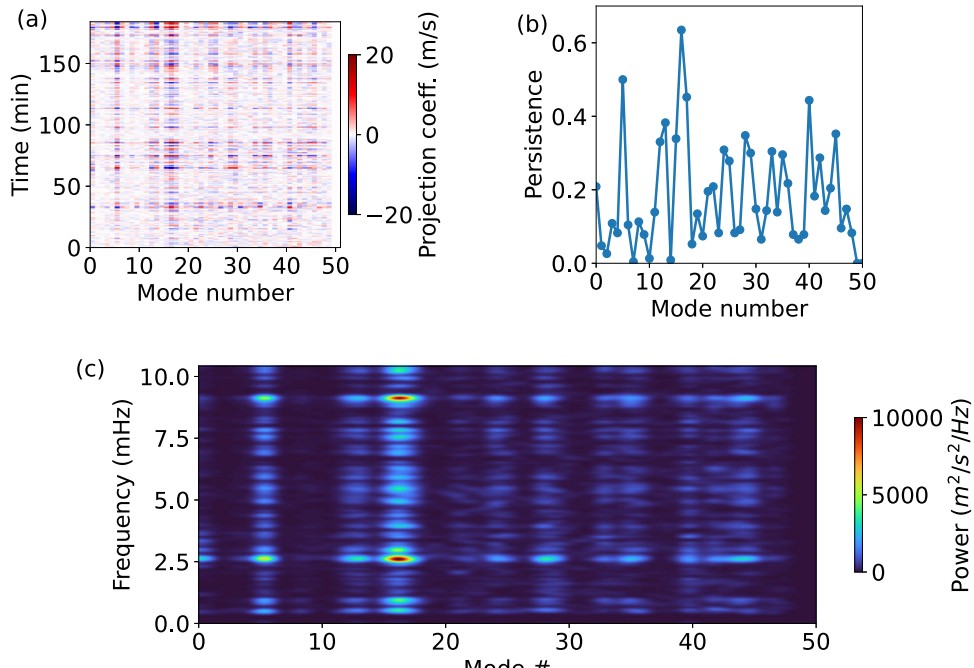

**Fig. 6 Temporal variation of the coefficients of the modal reconstruction. a** Temporal evolution of the coefficients of the modal reconstruction. The modes of the modal basis used are shown in Fig. 4. **b** Persistence of the eigenmodes oscillation as a fraction of the total duration of the observation. Some of the modes (e.g., n. 16) display a longer persistence with respect to others. **c** Power spectrum of the variation of the projection coefficients of the modes. The spectra are very similar for all the eigenmodes. Source data are provided as a Source Data file.

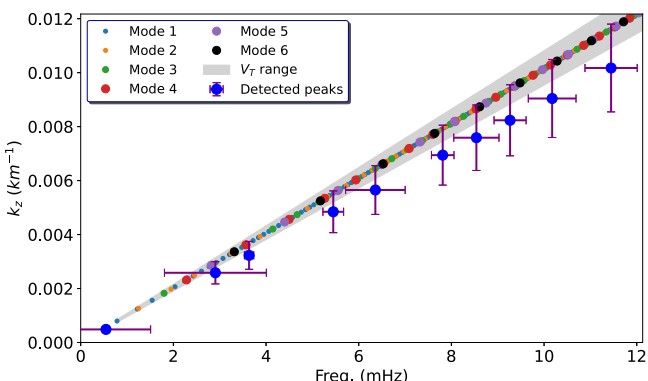

**Fig. 7 Dispersion relations.** Dispersion relation from the theoretical MHD model (blue dots) and from the data (orange dots). The gray shaded area represents the possible range of wavelengths, in the model, which are due to the range of possible values for the tube speed. The horizontal bars of the experimental data represent the width of the spectral features, and the vertical bars are given by the uncertainty on the height of formation of the line. Source data are provided as a Source Data file.

in the observational data, since theoretically they are a continuum[2]. In order to compare the dispersion relation obtained from the model with the data, we estimated the phase speed from a wavelet phase lag analysis of the velocity data at two different heights (i.e. photosphere and chromosphere), using simultaneous observations in the chromospheric Ca 8542 Å spectral line. This analysis was based on the well-known wavelet software by Torrence et al.[49]. In order to ensure the reliability of the phase speed estimate, we restricted ourselves to the 5 mHz band, which is the dominant frequency in the chromosphere, and only considered the phase measurements with a coherence of at least 70%. The PDF of the phase is shown in Fig. 9 where we see a distinct component around 2 radians. Assuming a height of formation of the chromospheric line Ca 8542 Å in the range 800–1000 km[50], and a formation height for the Doppler velocity signal of the Fe 6173 Å of 200 km[51], we obtain a phase speed in the range 5.6–7.5 km/s.

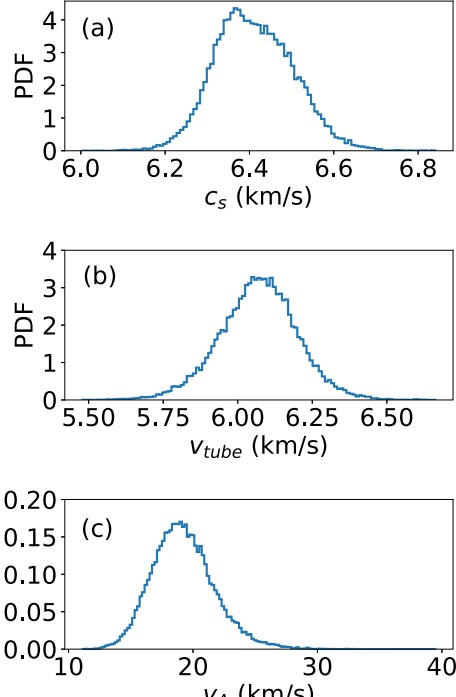

**Fig. 8 Characteristic speeds inside the sunspot.** Probability density function (PDF) of the main characteristic speeds inferred from a modeling based on spectropolarimetric inversions of the Fe 6173 Å spectral data within the umbra. **a** Sound speed in the umbra. **b** Tube speed in the umbra. **c** Alfvén speed in the umbra. Source data are provided as a Source Data file.

From this estimate we obtained the dispersion relation which is plotted in Fig. 7. Note that within the uncertainties of the two dispersion relations, which are obtained independently from both the model and the data, they are in very good agreement, with the caveat that the phase speed estimated from the observed

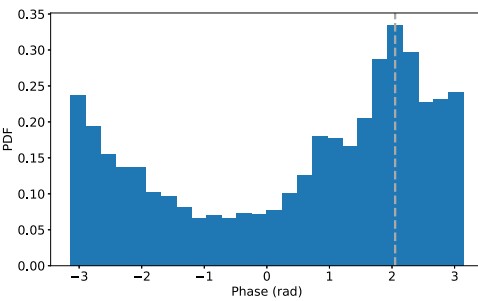

**Fig. 9 Phase lag of the wave signals between photosphere and chromosphere.** Probability density function (PDF) of the most coherent phase measurements at 5 mHz between the Doppler velocity signals sampled simultaneously at the two heights by analyzing the Fe 617.3 nm photospheric and the Ca 8542 Å chromospheric lines. The vertical dashed line marks the peak considered in the analysis. Source data are provided as a Source Data file.

phase lag is representative of the whole range of heights explored. This may explain the slightly different slope with respect to the theoretical model.

## Data availability

The data that support the findings of this study are available in repository IBIS-Archive (http://ibis.oa-roma.inaf.it/IBISA) and the SDO/AIA database (https://sdo.gsfc.nasa.gov/data/aiahmi/). The datasets generated during and/or analyzed during the current study are available from the corresponding author on request. Source data are provided with this paper.

## Code availability

This work made use of standard routines contained in the Matplotlib, Scipy and Numpy Python modules. The modal basis was computed using the MATLAB *delsq* and *eigs* routines. Python wavelet software provided by Evgeniya Predybaylo based on Torrence and Compo (1998) and is available at http://atoc.colorado.edu/research/wavelets/. VFISV code is freely available and can be downloaded from https://gitlab.leibniz-kis.de/borrero. The NICOLE spectropolarimetric inversion code is available at https://github.com/hsocasnavarro/NICOLE

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

## Acknowledgements

The authors wish to acknowledge scientific discussions with the Waves in the Lower Solar Atmosphere (WaLSA; www.WaLSA.team) team, which is supported by the Research Council of Norway (project number 262622), and The Royal Society through the award of funding to host the Theo Murphy Discussion Meeting "High resolution wave dynamics in the lower solar atmosphere" (grant Hooke18b/SCTM). M.S. acknowledges Doug Gilliam for his valuable support during the observing campaign at DST. Data analysis and visualization made use of the Scipy, Numpy, and Matplotlib Python libraries. This work is supported by The Royal Society, International Exchanges Scheme, collaboration with Chile No. IE170301 (V.F. and G.V.) and Brazil No. IES\R1\191114 (V.F. and G.V.), Science and Technology Facilities Council (STFC) grant No. ST/V000977/1, the Deanship of Scientific Research (DSR), King Faisal University, Al-Hassa, KSA for the financial support under Nasher Track No. 186354 (A.A.A.), the Invest NI and Randox Laboratories Ltd. for the award of a Research & Development Grant No. 059RDEN-1 (D.B.J.), the UK Science and Technology Facilities Council (STFC) grant No. ST/T00021X/1 (D.B.J.), the European Research Council under the European Union Horizon 2020 research and innovation program No. 682462 (S.J.), the Research Council of Norway through its Centres of Excellence scheme No. 262622 (S.J.), the STFC consolidated grant No. ST/S000240/1 (D.B.), the European Union's Horizon 2020 Research and Innovation program under grant agreement No. 82135 and No. 814335 (SOLARNET) and No 739500 (PRE-EST) (I.E., M.M., F.G.). This publication is part of the R+D+i project PID2020-112791GB-I00 financed by MCIN/AEI/10.13039/501100011033 (R.S. and J.T.). C.D.M. would like to thank the Northern Ireland Department for the Economy for the award of a PhD studentship. This research has made use of the IBIS-A archive.

## Author contributions

M.S. carried out the experiment and conceived and designed the study with G.V., V.F., A.A.A. and M.M. F.G. performed the data reduction with assistance from M.S., M.M., S.J. and I.E. M.S. performed scientific data analysis with support from S.J. and M.M. G.V., A.A.A. and V.F. conceived the numerical modeling. M.S., G.V., V.F., A.A.A., J.T., R.S., B.F. and D.B.J. interpreted and discussed the results with additional contributions from P.H.K., C.D.M.B., M.M., M.S., G.V. and V.F. drafted the manuscript with contributions from all coauthors. All authors read and approved the manuscript.

## Competing interests

The authors declare no competing interest.

## Additional information



