## [Peer Review File · Nature Communications]

REVIEWER COMMENTS

Reviewer #1 (Remarks to the Author):

The claim is made on the first detection of multiple resonant eigenmodes in a large scale magnetic flux structure on the solar surface. This conclusion is reached after the identification by spectral power analysis of multiple harmonics within the umbra of a sunspot, which are absent in the penumbra, and on the agreement between the spatial distribution of measured Doppler velocity fluctuations and the reconstruction of the signal from a combination of magnetohydrodynamic eigenmodes.

These results are of interest to the solar physics community in a broad context. Several aspect of the analysis require clarification. The importance and further implications of the results are insufficiently addressed.

A) Aspects that require clarification:

- Line 1 "... the presence of several simultaneous resonant modes in solar magnetic structures was never identified". Multiple harmonics have been detected in magnetic structures of the solar atmosphere. Later on the manuscript specifies that the study deals with "solar magnetic features like sunspots", but the initial general statement can lead to confusion.

- "MHD waves are thought to play a major role in heating the outer regions of the atmosphere to million-degree temperatures..." A recent review paper summarises the current state of the art (Van Doorselaere et al. 2020, 2020SSRv..216..140).

- On page 2, the claim is made that the intense magnetic field supresses the 5-minute oscillations associated to the global acoustic resonances of p-modes. According to Figure 2 power at 3 mHz is partly reduced but not wholly eliminated.

- In figure 2, no conclusion is reached concerning the separation in frequency of the multiple harmonics present in the umbra. Is the spectrum uniformly or non-uniformly distributed? A non-uniformly distributed spectrum may be indicative of structuring with important information on it. Do the authors have an hypothesis for the spacing in frequency of the power spectrum in the umbra?

Does the reconstructed magnetohydrodynamic solution show a similar spectral density distribution?

- The umbral penumbral boundary in the B-w diagram is delimited by the magnetic field strength. This implicitly means the level of inhomogeneity in the modeling should at least be 1D, i.e., $B = B(r)$ in the sunspot. In the eigenmode analysis the uniformity/inhomogeneity of the adopted equilibrium model is not discussed.

- In Chapter 6 of Roberts (2019, MHD waves in the solar atmosphere, Cambridge University Press), Eqs. (6.29) and (6.22) govern the pressure and vertical velocity perturbations in magnetic flux tubes with $B=B(r)$. These equations are applicable to both photospheric tubes and to coronal tubes, provided the correct ordering of phase speeds is considered. It is unclear to me how these equations are translated into the simple governing equation adopted here. Besides the change from cylindrical to Cartesian geometry, one would need to assume a uniform equilibrium, which is in contradiction with figure 2 and the fact that the magnetic field is highly inhomogeneous in sunspots. Also, the study seems to identify the oscillations as sausage slow body modes, with rotational symmetry about the tube axis, and the modelled solutions are highly non-axisymmetric.

- The spatial distribution of the solutions seems to be determined to a large extent by the cross-sectional shape of the waveguide by imposing boundary conditions at the u-p boundary. This boundary is defined by the observed shape which is highly non-axisymmetric. How this is implemented is not explained in detail. It would seem to me that the spatial distribution of the velocity perturbations is entirely determined by the irregular boundary conditions and not by the structuring of the medium.

- A method was presented by Albidah et al. (2021, 2021RSPTA.37900181A) to identify the dominant MHD modes in data such as the ones considered here. The authors could comment on the similarities/differences, advantages/disadvantages of both methods.

- The measured Doppler velocity is projected onto the computed modal basis to reconstruct the observation by a linear combination of eigenmodes with different energy. It is not clear what energy means in this context and what makes those 9 the chosen ones. Is there any correspondence with known methods such as Principal Component Analysis, Complex Empirical Orthogonal Function, or Empirical Mode Decomposition?

B) Importance of results and further implications

In the opening highlighted text, the manuscript states that "Our findings provide the detection of such peculiar oscillatory pattern in large sunspots" and that these results "open a new avenue, identifying a methodology to examine modes in intense solar magnetic structures for new diagnostic applications". At the end of the main text, the authors conclude that "...the present analysis not only provides the first evidence of the co-existence of many MHD modes and their radial overtones in an umbra, but also paves the way to use this methodology to advance the field of local magnetoseismology"

Sunspots are definitely among the best studied structures in solar physics. Nowadays, imaging, spectroscopic and spectropolarimetric observations enable us to measure in high resolution many physical parameters. Helioseismic tomography, spectropolarimetric inversions and numerical modelling have greatly advanced our understanding of the complex 3D field and plasma structure and flows at different optical depths. Confirmation that sunspot umbrae can harbour multiple MHD eigenmodes is an important result. However, the manuscript does not explain in detail the reasons why and the ways in which the presented methodology offers an added value or better information in comparison to those other current methods, in particular when exceptionally intense sunspots seem to be required.

Reviewer #2 (Remarks to the Author):

The authors analyze the oscillations measured in a remarkable data set, with an uncommonly long temporal series of an uncommonly large sunspot. Velocity fluctuations are interpreted as the contribution of several resonant wave modes supported by the sunspot magnetic structure. The analysis and results go in line with those previously reported by Fujimura & Tsuneta, *ApJ*, 702, 144 (2009), Morton et al., *ApJL*, 729, L18 (2011), Moreels et al., *A&A*, 555, A75 (2013), Jafarzadeh et al., *ApJS*, 229, 10 (2017), Keys et al., *ApJ*, 857, 28 (2018) for small magnetic flux tubes and Jess et al., *ApJ*, 842, 57 (2017) for sunspots. The paper shows an interesting work and it should be well received by the community. However, I unfortunately do not think it possesses sufficient scientific significance and novelty. Therefore, I do not recommend this manuscript to be accepted for publication in *Nature Communications*, but encourage the authors to submit it to a different journal.

I have the following major comments that the authors may want to consider for a future submission:

1. The analysis illustrated in the manuscript focus on the characterization of the velocity fluctuations measured at a single time step as the superposition of a set of eigenmodes. The results are

expanded in the associated movie, showing the results obtained for a certain temporal span. I understand the observed velocity has been individually projected onto the computed modal basis for each time step. This approach is not sufficient to claim the detection of the resonant modes. Any snapshot of the velocity field (even if it is not originated by resonant modes) can be expressed in terms of the computed basis with reasonable success. To confirm the existence of those resonant modes, the model (a selection of modes, coefficients, and their eigenfrequencies) should consistently explain not only the instantaneous velocity field but also its temporal evolution, at least during some temporal range while the oscillations remain coherent. The projection coefficients for most of the time steps showed in the movie exhibit some preference for certain modes (1, ~15-17). I believe this is an indication of the existence of those resonances. However, the modeling of the temporal evolution must be discussed in the manuscript.

2. The manuscript lacks the discussion of the results in the context of other recent works:

2.1. Jess et al., *ApJ*, 842, 59 (2017) detected $m=1$ mode waves in observations of an umbral chromosphere. The last paragraph from the first page of the manuscript mentions the detection of single eigenmodes in small-scale magnetic structures but overlooks this more related result. In the present manuscript, the authors find a large contribution from a sausage-like mode. Numerical works have found that the excitation of $m=0$ and $m=1$ modes strongly depends on the radius of the flux tube. The kink mode is the dominant mode in tubes with a small radius, while the sausage mode is dominant for large tubes (Daifallah et al. *SoPh*, 268, 309, 2011). These numerical results qualitatively agree with the preference for the sausage mode in a big sunspot such as that analyzed in the manuscript.

2.2. Recent studies have found evidences of a chromospheric resonant cavity in the umbral chromosphere (Jess et al., *Nature Astronomy*, 4, 220, 2020; Felipe et al. *ApJ*, 900, 29, 2020). What is the relation between resonances produced by reflections at the transition region and those created by the magnetic structure?

2.3. The same sunspot data was analyzed by Stangalini et al., *ApJ*, 869, 110 (2018). They focused on the study of circular polarization oscillations, which were interpreted as a propagating surface mode localized in the umbra-penumbra boundary. This work should be cited and discussed in the manuscript.

The following are my minor comments:

3. In the second paragraph from page 3, the authors describe that, unlike many previous works, the modeling did not assume an idealized cylindrical shape, but the actual cross-sectional shape of the umbra was employed. Describe how this shape was defined (contour of constant intensity or contour with $B=1867$ G?).

4. In the same paragraph, the authors mention that the governing equation of the velocity perturbation was solved setting $v_z=0$ at the umbra-penumbra boundary. It is indicated that this choice was done to be consistent with the observational data. This claim contrasts with the analysis from Stangalini et al., ApJ, 869, 110 (2018) of the same data, who measured line-of-sight velocity fluctuations at the umbra-penumbra boundary with a clear and consistent phase lag with the magnetic perturbations. Those magnetic perturbations were associated with propagating waves.

5. Add references for the equations employed for the modeling of the MHD waveguide on page 3.

6. The figure number in the first paragraph from page 4 is wrong. I guess that in the sentence "results in the orthogonal modal basis shown in Fig. 3", authors refer to Fig. A2.

7. In the second paragraph from page 4, the sentence "Most of the energy is contained in the first 5-10 eigenmodes" refers to the first modes illustrated in Fig. A2 or to the 5-10 most energetic modes?

8. In the same paragraph, it is stated that "Both the fundamental and first "radial" overtone of this mode have the largest contribution." Explicitly indicate the specific modes from the basis illustrated in Fig. A2.

9. Related to comment #1. The single time step illustrated in the manuscript (figures 1 and A3) is better fitted with a selection of eigenmodes where the fundamental mode is the dominant. This is not the common situation, as shown in the associated movie. This needs to be discussed.

10. In figure A1, indicate the selected sound speed value.

11. In the first paragraph from page 8, correct the color of the line. Orange instead of yellow.

12. In figure A2, the blue color means downward according to the chosen sign convention. I would recommend using blue color for upward since this is the general convention. Also, I understand that in figure 1 negative Doppler velocities (blue) correspond to upward.

13. Indicate the time in both movies.

Reviewer #3 (Remarks to the Author):

Using IBIS observations of one of the largest sunspots of recent years, the authors have obtained complex time series of its umbral oscillations over 3 hrs and sought to model them in terms of normal modes described by standard MHD theory. The results are broadly consistent, and on this basis the authors conclude that the standard theory gives an adequate and useful description of umbral waves.

There are several reasons for measuring and understanding umbral oscillations. One is to place seismic constraints on the important and mysterious subsurface structure, including dynamo action. This is extremely difficult, and the authors do not discuss it. Another is to provide boundary conditions for waves travelling from the umbra up to the chromosphere and beyond. Again, this is not discussed. Nevertheless, the presented results may find application in these areas.

I have a number of questions and comments.

1. On page 1, it is stated that sunspots are less subject to granular forcing due to their large sizes. However, this is more because of the suppression of convection by the strong magnetic fields.

2. Citing ref 18 on absorption of p-modes by sunspots, the authors say that p-mode absorption "appears as a natural explanation for the dominant (5-minute) power in sunspots". This is a misunderstanding of p-mode absorption. The absorption is actually a mode conversion process, at the layer where the sound and Alfvén speeds are equal, whereby the predominantly acoustic p-modes partially turn into magnetic waves that are either lost downward along the magnetic field lines into the solar interior, or upward as fast MHD waves into the chromosphere. In either case, energy is partially removed from those p-modes, and those that emerge from spots do so with

reduced amplitude. This has nothing to do with whether or not spots exhibit that same 5-minute period peak as the quiet Sun. Instead, it is a matter that these quiet Sun p-modes are global normal modes of the Sun's surface layers, with distinct eigenfrequencies. They propagate around the Sun continuously, including inevitably through sunspots (where they are partially absorbed), so of course they will show wave power at these dominant frequencies.

3. In Fig 2, is B the total magnetic field, or just its line-of-sight component, or just the vertical component? By the way, the eigenmode power peaks at 5.5 mHz and 9 mHz and $B=2500$ G are very interesting, and presumably tell us something.

4. On page 3: the vertical wavenumber k_z appears in m_0 in the wave equation. but little is said of it. That equation is actually only valid for tubes that are uniform along their lengths, the vertical direction in this instance. However, umbral magnetic field expands rapidly with height, so tube cross-section is certainly not uniform. More importantly though, the gravitational density scale height is only about 100 km, which is tiny in the context of a 30,000 km sunspot. So how is this equation relevant to umbrae. And if it is, what values of k_z have been used in the simulations and why?

5. The authors use a $v_z=0$ boundary condition at the umbra-penumbra boundary. It is by no means clear why this is appropriate. Indeed, it is not easy to see how it would be physically imposed by the Sun. Can the authors comment?

6. At the top of page 4, the authors refer to Fig 3. However, as far as I can see, there is no Fig 3.

7. Page 4, paragraph "The remarkable agreement ..." does not really go to the question of which frequencies are prominent in the umbra. The 5.5 mHz and 9 mHz frequencies seen in Fig 2 are restricted to near the umbral centre. So their eigenfunctions should presumably be strongly peaked there. But no data on this has been presented. Fig A2 could be made much more useful by labelling each panel by its eigenfrequency, rather than the mode number.

8. According to the top left panel in Fig A3, the retained modes are 1, 13, 14, 16, 17, 18, 29, 34 and 36. However, this differs from those distinguished by people frames in Fig A2 (1, 12, 13, 15, 16, 17, 28, 33 and 35). By the way, purple is a poor choice of highlight colour, as it is difficult to see. I suggest red or dark green may be better.

Dear referee,

We thank you very much for your report and for recognizing our work as being of potential interest to the community in a broad context. In the following we address your points one by one.

Here we would also like to comment on the possible implications of our research. We agree with the referee that this aspect is not sufficiently addressed in the text. Similarly to the role played by the p-modes in many fields, we believe that a methodology to routinely study the eigenmodes of magnetic structures down to the photospheric heights could have a number of key applications including the seismic constraining of the solar subsurface structure, as well as the assessment and modelling of the noise contribution given by MHD oscillations to the detection of exoplanets around magnetically-dominated stars. We believe that it is currently impossible to imagine all the possible applications, and in the paper we now try to list a few.

However, our results do not only propose a methodology for the study of resonant modes in magnetic structures, but primarily show how it was possible to detect and study for the first time how extremely large sunspots can host a rich superposition of MHD wave modes, even including high order oscillations. These are by far the largest scale MHD oscillations ever detected in the photosphere, and this provides constraints on the mechanisms of excitation of similar sunspot oscillations and their drivers as well.

Reviewer #1 (Remarks to the Author):

The claim is made on the first detection of multiple resonant eigenmodes in a large-scale magnetic flux structure on the solar surface. This conclusion is reached after the identification by spectral power analysis of multiple harmonics within the umbra of a sunspot, which are absent in the penumbra, and on the agreement between the spatial distribution of measured Doppler velocity fluctuations and the reconstruction of the signal from a combination of magnetohydrodynamic eigenmodes.

These results are of interest to the solar physics community in a broad context. Several aspects of the analysis require clarification. The importance and further implications of the results are insufficiently addressed.

A) Aspects that require clarification:

- Line 1 "... the presence of several simultaneous resonant modes in solar magnetic structures was never identified". Multiple harmonics have been detected in magnetic structures of the solar atmosphere. Later on the manuscript specifies that the study deals with "solar magnetic features like sunspots", but the initial general statement can lead to confusion.

Please note that we do not use the term "harmonics" at all in the manuscript since this might imply to the reader that there are standing modes in the vertical direction. With this data set we can make no claims about this. It is not clear to us if the referee means "harmonics" along the flux

tube axis direction or in the radial direction (assuming cylindrical geometry) or both. In the manuscript we simply label the modelled orthogonal slow body modes, physically constrained by the observed irregular cross-sectional shape, as numbers 1-50. For waves propagating along a magnetic flux tube, theoretically there can be a continuum of possible eigenfrequencies for each slow body mode. This scenario is actually much more consistent with what we observe. Observationally, each individual slow body mode has been shown to have a broadband of frequencies associated with it, suggesting a driver which is also broadband in ω . For the observed umbral shape the unique eigenvalue m_0 is numerically calculated for each mode. Physically, m_0 depends on the values of v_A , c_s , ω and k_z . From inversion estimates we get values for v_A and c_s . The analysed frequencies of interest range up to 10 mHz. Hence the unique value of k_z for each ω can be easily calculated, suggesting a driver which is also broadband in k_z . In the following plots we show the modelled phase speed vs k_z and frequency vs k_z . Since the slow body mode is very weakly dispersive, it can be seen in the frequency vs k_z plot, that all the modes when overplotted (in the frequency range up to 10 mHz) are practically sitting on a straight line.

In the abstract we refer to the coexistence of tens of such slow body modes which are simultaneously excited in the same umbral magnetic flux tube. This is the first time this behaviour is seen in such a large umbral magnetic structure but also in small-scale structures where generally just one or two modes are detected (e.g. the fundamental radial kink and sausage

modes). This is the novel aspect of the current work. However, to avoid confusion, we have modified the text.

- "MHD waves are thought to play a major role in heating the outer regions of the atmosphere to million-degree temperatures..." A recent review paper summarises the current state of the art (Van Doorselaere et al. 2020, 2020SSRv..216..140).

Thanks, we have added this reference.

- On page 2, the claim is made that the intense magnetic field suppresses the 5-minute oscillations associated to the global acoustic resonances of p-modes. According to Figure 2 power at 3 mHz is partly reduced but not wholly eliminated.

Correct, we agree. Text rephrased.

- In figure 2, no conclusion is reached concerning the separation in frequency of the multiple harmonics present in the umbra. Is the spectrum uniformly or non-uniformly distributed? A non-uniformly distributed spectrum may be indicative of structuring with important information on it. Do the authors have an hypothesis for the spacing in frequency of the power spectrum in the umbra? Does the reconstructed magnetohydrodynamic solution show a similar spectral density distribution?

Figure 2 was primarily to show how remarkably different the frequency spectrum of Doppler oscillations are inside and outside of the umbra. Actually explaining why the frequency spectrum has the structure it has is a very challenging open question. To further illustrate the difficulty in doing this below we show that each individual slow body mode has its own broadband frequency range and hence a broadband k_z . Therefore, whatever is driving all these oscillations in the umbra is both broadband in ω and k_z . However, it can also be seen that there is a complex and non-trivial fine structuring to the power distribution for each mode.

We know of no simulated sunspot that has been realistic enough to produce such a rich superposition of wave modes and such a finely structured frequency spectrum. E.g. Daifallah et al. (SoPh, 268, 309, 2011) using the SLiM (Simulation of Wave Propagation through an Inhomogeneous, Magnetised Solar Atmosphere) code only excited the fundamental radial slow body sausage and kink waves in a simplified sunspot (no umbra/penumbra structuring) using a

Gaussian f-mode packet. However, the magnetic flux tube was embedded in a stationary atmosphere (no convection) and the cross-sectional shape was perfectly circular. So currently we do not have the advanced numerical modelling required to even begin to start explaining the basic structure of the B-omega diagram in the umbral region or, on an even finer scale, the individual frequency power spectrum of each mode. Hence, that is why we have not hypothesized anything about the B-omega diagram. From the model employed in this work we can at most calculate the k_z associated with each omega for each mode, using estimates v_A and c_s from inversions. It cannot begin to predict omega-k output of the system since the sub-surface input is unknown.

- The umbral penumbral boundary in the B-w diagram is delimited by the magnetic field strength. This implicitly means the level of inhomogeneity in the modeling should at least be 1D, i.e., $B = B(r)$ in the sunspot. In the eigenmode analysis the uniformity/inhomogeneity of the adopted equilibrium model is not discussed.

As correctly said in one of the following points of the report, the spatial distribution of the solutions is determined to a large extent by the cross-sectional shape of the waveguide.

However, in the new version of the draft we have added the sound and Alfvén speed maps obtained from spectropolarimetric inversions using the Nicole code (Socas Navarro et al. 2014). Despite these maps, that we also show below for simplicity, are obtained at a constant tau not a geometrical height, they do not show a significant variation of the two quantities in the umbra, but only a strong discontinuity at the umbra-penumbra boundary which can be ascribed to the variation of the geometrical height of the response function of the spectral line.

For comparison in the maps we also plot the contour considered for the calculations of the eigenmodes of the umbra and area at the center of the umbra which is affected a saturation of the magnetic sensitivity due to the saturation of the Zeeman splitting of the Fe I 617.3 nm spectral line.

It is worth underlining that the nearly constant value of the Alfvén and sound speed in the umbra are probably due to the complete absence of residual convection and umbra dots within the umbra itself. This is not generally the case in sunspots and is probably due to the intensity of the magnetic field strength in this case.

- In Chapter 6 of Roberts (2019, MHD waves in the solar atmosphere, Cambridge University Press), Eqs. (6.29) and (6.22) govern the pressure and vertical velocity perturbations in magnetic flux tubes with $B=B(r)$. These equations are applicable to both photospheric tubes and to coronal tubes, provided the correct ordering of phase speeds is considered. It is unclear to me how these equations are translated into the simple governing equation adopted here. Besides the change from cylindrical to Cartesian geometry, one would need to assume a uniform equilibrium, which is in contradiction with figure 2 and the fact that the magnetic field is highly inhomogeneous in sunspots. Also, the study seems to identify the oscillations as sausage slow body modes, with rotational symmetry about the tube axis, and the modelled solutions are highly non-axisymmetric.

We would like to stress that we did not start our modelling with cylindrical geometry, and all numerical modelling was performed in the Cartesian coordinate system. Due to the irregular shape of the umbra there would be no advantage in using cylindrical coordinates to solve the eigenvalue problem numerically. The governing Helmholtz type equation was derived from the ideal linearized MHD equations assuming $v_z(x,y)$ and a uniform background V_a, C_s . Also, to be consistent with the observation the v_z perturbations were assumed "0" at the boundary, indicative of slow body modes being observed. As can be seen in the maps above, the inhomogeneities are mainly located outside the umbra and no sign of residual convection or umbral dots found to testify the presence of large inhomogeneities. This can be also seen in the core intensity map where no umbral dot or residual convection can be observed.

Also, the study seems to identify the oscillations as sausage slow body modes, with rotational symmetry about the tube axis, and the modelled solutions are highly non-axisymmetric.

The analysed sunspot is not completely axially symmetric (but in a first approximation it can be taken so) but the breaking of symmetry is particularly evidenced in the high order modes i.e. by the modelled solutions.

The structure of the modelled modes follows the shape of the umbra, as is physically expected (see Fig.1).

- The spatial distribution of the solutions seems to be determined to a large extent by the cross-sectional shape of the waveguide by imposing boundary conditions at the u-p boundary. This boundary is defined by the observed shape which is highly non-axisymmetric. How this is implemented is not explained in detail. It would seem to me that the spatial distribution of the velocity perturbations is entirely determined by the irregular boundary conditions and not by the structuring of the medium.

Yes, indeed. As stated in the text, the irregular shape of the boundary is an essential point in our modelling. As can be also seen in Fig. 1, and recognized by the referee, the spatial distribution of the velocity perturbations is predominantly determined by the irregular shape of the boundary. Of course, the structuring of the medium could also influence eigenfunctions, but as shown by the inversions this would be mainly at the boundary regions with higher order modes which have spatial structuring at a similar length scale to this inhomogeneity. This does not concern the most dominant modes which we have identified here.

- A method was presented by Albidah et al. (2021, 2021RSPTA.37900181A) to identify the dominant MHD modes in data such as the ones considered here. The authors could comment on the similarities/differences, advantages/disadvantages of both methods.

We have added a specific discussion on that paper. Ours is a decomposition using a pre-computed modal basis, in the same way, FFT, wavelet, PCA, DMD work. This is not a simulation from which to expect a “temporal evolution”.

However, in contrast to the aforementioned methods, in our case, the modal basis is not defined from e.g. a statistical point of view as in the case of PCA or DMD. On the contrary, our basis is constructed on physical assumptions/expectations (i.e., a linear ideal MHD oscillation, governed by a Helmholtz type equation with the actual cross-sectional shape of the boundary from the observation) which is, in principle, not dependent on the data. The fact that these physical assumptions provide a good representation of the actual resonant modes is remarkable. This conclusion could not be drawn from PCA or other decomposition methods because their modal basis does not rely on physical assumptions.

In the following we plot the total variance as a function of the number of modes used in the reconstruction. This is done for both a normal ordering of the modal basis and for an ordering based on the variance of each mode (i.e. the fraction of information of the original signal that can be described by each single mode).

Here two things are worth noting. First, some modes (e.g. mode number 15, 16 and 17) account for a larger amount of total variance of the original signal with respect to others. Second, as can be seen for the reconstruction using the modal basis sorted by variance, adding more modes than 20-25 does not improve the reconstruction, meaning that these additional modes, not being representative of the signal, cannot describe it. Indeed, their projection coefficient is very small as they are not able to represent part of the variance of the signal.

However, we note here that by using a limited number of modes we are able to account for a significant fraction of the total variance of the signal, meaning that these modes, even if independently calculated, provide a good representation of the signal itself.

We underline that, unlike other methodologies (e.g. PCA, EMD,...), although our modal basis is computed upon *a priori* physical assumptions, it is able to reconstruct the signal and this is a validation of the physical assumptions themselves.

In our opinion it is a remarkable fact that by primarily considering the actual shape of the umbra for the computation of the eigenmodes we are able to reconstruct, with great accuracy, the complicated velocity signal observed on the Sun.

- The measured Doppler velocity is projected onto the computed modal basis to reconstruct the observation by a linear combination of eigenmodes with different energy. It is not clear what energy means in this context and what makes those 9 the chosen ones. Is there any correspondence with known methods such as Principal Component Analysis, Complex Empirical Orthogonal Function, or Empirical Mode Decomposition?

By energy we mean the fraction of the total variance a particular eigenmode represents. We have clarified this in the text. Our approach is very similar to PCA reconstruction, but with one simple yet fundamental difference. The modal basis is not computed from the signal itself, but it is based on physical assumptions, i.e., that the oscillations are governed by the linear ideal MHD equations and the eigenfunctions are physically constrained by the actual observed irregular boundary shape. It is not always the case that PCA modes correspond to physical ones, in fact PCA will give as many modes as there are time snapshots in the data set. As for the other methods mentioned, as their name suggests the modal bases are “empirically” constructed from the signal, not necessarily what is physically expected. We have now mentioned these points in the paper.

B) Importance of results and further implications

In the opening highlighted text, the manuscript states that "Our findings provide the detection of such peculiar oscillatory pattern in large sunspots" and that these results "open a new avenue, identifying a methodology to examine modes in intense solar magnetic structures for new diagnostic applications". At the end of the main text, the authors conclude that "...the present analysis not only provides the first evidence of the co-existence of many MHD modes and their radial overtones in an umbra, but also paves the way to use this methodology to advance the field of local magnetoseismology"

Sunspots are definitely among the best studied structures in solar physics. Nowadays, imaging, spectroscopic and spectropolarimetric observations enable us to measure in high resolution many physical parameters. Helioseismic tomography, spectropolarimetric inversions and numerical modelling have greatly advanced our understanding of the complex 3D field and plasma structure and flows at different optical depths. Confirmation that sunspot umbrae can harbour multiple MHD eigenmodes is an important result. However, the manuscript does not explain in detail the reasons why and the ways in which the presented methodology offers an added value or better information in comparison to those other current methods, in particular when exceptionally intense sunspots seem to be required.

Elucidating the impact our results may have in other contexts or applications is an important suggestion.

To address this point we believe it is appropriate to start with what said by the third referee: *“There are several reasons for measuring and understanding umbral oscillations. One is to place seismic constraints on the important and mysterious subsurface structure, including dynamo action. This is extremely difficult, and the authors do not discuss it. Another is to provide boundary conditions for waves travelling from the umbra up to the chromosphere and beyond.”*

Similarly to what is done with global resonances (e.g., p-modes) of stars, which are nowadays commonly employed to study the interior of the Sun (and other stars!), the study of the global resonances of magnetic structures such as sunspots or star spots can provide crucial information about a magnetically dominated atmosphere.

Coronal seismology has been already proposed (Roberts *et al.* 1984 *Astrophys. J.* 279, 857) and applied as a diagnostic tool for determining the physical conditions of the coronal environment. However, nothing similar was possible in the lower solar atmosphere where, as our results suggest, resonant oscillations are hidden by large amplitude incoherent fluctuations due to, e.g., p-modes leakage.

Besides, the study of global resonances at photospheric heights has an additional advantage. If applied to a statistically significant number of magnetic structures, our approach may offer the unique possibility to study how magnetic structures respond to the sub-surface drivers and therefore provide the basis with which to investigate how magnetic fields govern the outer layers of the Sun's atmosphere.

In addition, we would like to stress one point. Our paper does not only present a new methodology but reports on the largest coherent MHD oscillations ever observed in the solar photosphere so far. This aspect has a number of implications on the drivers responsible for that and its broadband omega and k structure. This observation is also unique for a number of other reasons, e.g., it is the first time that a sunspot with this dimension has been observed at very high spatial, temporal and spectral resolution and this allowed not only the discovery of remarkably unique high order MHD oscillations in it, but also offers the possibility for testing new methodologies for further investigation.

Dear referee,

We thank you for the comments and for recognizing our work as potentially interesting for the community.

In what follows we address these points one by one.

Reviewer #2 (Remarks to the Author):

The authors analyze the oscillations measured in a remarkable data set, with an uncommonly long temporal series of an uncommonly large sunspot. Velocity fluctuations are interpreted as the contribution of several resonant wave modes supported by the sunspot magnetic structure. The analysis and results go in line with those previously reported by Fujimura & Tsuneta, ApJ, 702, 144 (2009), Morton et al., ApJL, 729, L18 (2011), Moreels et al., A&A, 555, A75 (2013), Jafarzadeh et al., ApJS, 229, 10 (2017), Keys et al., ApJ, 857, 28 (2018) for small magnetic flux tubes and Jess et al., ApJ, 842, 57 (2017) for sunspots. The paper shows an interesting work and it should be well received by the community. However, I unfortunately do not think it possesses sufficient scientific significance and novelty. Therefore, I do not recommend this manuscript to be accepted for publication in Nature Communications, but encourage the authors to submit it to a different journal.

We appreciate the effort of the community in identifying eigenmodes in various solar magnetic structures. However, we would like to stress that none of the papers mentioned here report such a rich superposition of MHD wave modes, including high order oscillations over similar extremely large spatial scales. Indeed, they only report on the presence of fundamental radial sausage or kink modes in the best cases and mostly in small scale magnetic structures (a few Mm maximum). The current work reports the very first time that such a large sunspot has been observed at these high spatial and temporal resolutions and allows us to validate a new technique for eigenmode decomposition. There is no other data set like this and it is the first time that such a large umbral magnetic structure in the Sun was observed with this accuracy.

I have the following major comments that the authors may want to consider for a future submission:

1. The analysis illustrated in the manuscript focus on the characterization of the velocity fluctuations measured at a single time step as the superposition of a set of eigenmodes. The results are expanded in the associated movie, showing the results obtained for a certain temporal span. I understand the observed velocity has been individually projected onto the computed modal basis for each time step. This approach is not sufficient to claim the detection of the resonant modes. Any snapshot of the velocity field (even if it is not originated by resonant modes) can be expressed in terms of the computed basis with reasonable success. To confirm the existence of those resonant modes, the model (a selection of modes, coefficients, and their eigenfrequencies) should consistently explain not only the instantaneous velocity field but also its temporal evolution, at least during some temporal range while the oscillations remain coherent. The projection coefficients for most of the time

steps showed in the movie exhibit some preference for certain modes (1, ~15-17). I believe this is an indication of the existence of those resonances. However, the modelling of the temporal evolution must be discussed in the manuscript.

The temporal evolution of the modal projection coefficients has been added to the paper. Here we report the same plot added to the text where it is seen several modes have a significant persistence (i.e. the fraction of total time their amplitude is above the threshold of 7 m/s) As can be seen in the diagram, there is a conspicuous modulation of the projection coefficients typical of a non-stationary signal as reported many times for MHD waves in solar magnetic structures. This long term modulation is in phase for the most significant modes.

2. The manuscript lacks the discussion of the results in the context of other recent works:

2.1. Jess et al., ApJ, 842, 59 (2017) detected $m=1$ mode waves in observations of an umbral chromosphere. The last paragraph from the first page of the manuscript mentions the detection of single eigenmodes in small-scale magnetic structures but overlooks this more related result. In the present manuscript, the authors find a large contribution from a sausage-like mode. Numerical works have found that the excitation of $m=0$ and $m=1$ modes strongly depends on the radius of the flux tube. The kink mode is the dominant mode in tubes with a small radius, while the sausage mode is dominant for large tubes (Daifallah et al. SoPh, 268, 309, 2011). These numerical results qualitatively agree with the preference for the sausage mode in a big sunspot such as that analyzed in the manuscript.

We are aware of these papers however as stated in the text only single eigenmodes were detected so far and in comparably much smaller magnetic flux tubes. Here we would like to stress that this is the largest sunspot ever observed at this spatial and temporal resolution (i.e. our observations are acquired at a spatial resolution which is at least a factor of 5 better than HMI). With regard to the Daifallah et al. paper, please note that what they refer to as "large tube" is at least ten times smaller than the diameter of the umbra of our sunspot.

What we observe in this huge structure is the simultaneous excitation of many eigenmodes. There are only very few sunspots like this in one solar cycle and as far as we know none of them has ever been observed with this high resolution.

The fact that so many modes are excited in such a structure is, in our opinion, already an interesting result in itself. None of the magnetic structures analysed in previous works can be compared to this. We have further commented on this and included more references as suggested.

2.2. Recent studies have found evidences of a chromospheric resonant cavity in the umbral chromosphere (Jess et al., Nature Astronomy, 4, 220, 2020; Felipe et al. ApJ, 900, 29, 2020). What is the relation between resonances produced by reflections at the transition region and those created by the magnetic structure?

This is an interesting and still open question in our opinion. However, we are not able with the current data to address this point.

2.3. The same sunspot data was analyzed by Stangalini et al., ApJ, 869, 110 (2018). They focused on the study of circular polarization oscillations, which were interpreted as a propagating surface mode localized in the umbra-penumbra boundary. This work should be cited and discussed in the manuscript.

We have now cited this paper and have pointed out that a propagating surface mode was also identified in the same sunspot. This is further confirming the simultaneous excitation of many modes in the same structure as indicated by our results. However, in our case we detect several body modes with high radial order that coexist in the umbra.

The following are my minor comments:

3. In the second paragraph from page 3, the authors describe that, unlike many previous works, the modeling did not assume an idealized cylindrical shape, but the actual cross-sectional shape of the umbra was employed. Describe how this shape was defined (contour of constant intensity or contour with $B=1867$ G?).

The shape was estimated by considering the $v=0$ boundary. Incidentally this is quite close to the $B=1867$ G boundary but not completely the same. In our test we have actually used both and found the resulting modal basis being more representative of the actual velocity pattern.

4. In the same paragraph, the authors mention that the governing equation of the velocity perturbation was solved by setting $v_z=0$ at the umbra-penumbra boundary. It is indicated that this choice was done to be consistent with the observational data. This claim contrasts with the analysis from Stangalini et al., ApJ, 869, 110 (2018) of the same data, who measured line-of-sight velocity fluctuations at the umbra-penumbra boundary with a clear and consistent phase lag with the magnetic perturbations. Those magnetic perturbations were associated with propagating waves.

Please note that the mentioned papers deal with fluctuations not the mean velocity at the boundary. The observation shows that v_z oscillations decay very rapidly at the umbra-penumbra boundary. Theoretically, this is expected for a slow body mode, even if the environment is quiescent, which is certainly not the case for an actual umbra. We have used the $v_z=0$ boundary of the k - ω filtered data which is practically very close to the $B=1867$ G found in the literature. Had the observed modes been more consistent with surface modes then the $v_z=0$ boundary condition would not be valid since these modes have their maximum amplitude at the boundary. Some possible reasons we only detect slow body modes may be due to the nature of the driver, or perhaps there is an inherent "rigidity" of the umbra-penumbra boundary which prevents the excitation of surface modes, however at this stage this is purely speculation.

5. Add references for the equations employed for the modeling of the MHD waveguide on page 3.

To model the basis eigenfunctions on which to project the observed signal we simply use ideal linearized MHD equations in Cartesian coordinates. To account for the irregular cross-sectional shape of the umbra we assumed a linear perturbation in $v_z(x,y)$ and that the background v_A and c_s were not changing to derive the governing Helmholtz type equation. We have clarified these points in the text.

6. The figure number in the first paragraph from page 4 is wrong. I guess that in the sentence "results in the orthogonal modal basis shown in Fig. 3", authors refer to Fig. A2.

Yes, thanks. Fixed.

7. In the second paragraph from page 4, the sentence "Most of the energy is contained in the first 5-10 eigenmodes" refers to the first modes illustrated in Fig. A2 or to the 5-10 most energetic modes?

We have changed this part also in consideration of a revised plot (A3) where we now show the reconstruction accuracy as a function of the number of eigenmodes reordered by their variance. In this case it is more clear that, although most of the energy is contained in the first 10-15 modes, a plateau in the reconstruction accuracy is only reached after using 30-35 modes, while the rest, modes 10-15 are not able to reproduce the signal thus do not contribute to improving the reconstruction.

8. In the same paragraph, it is stated that "Both the fundamental and first "radial" overtone of this mode have the largest contribution." Explicitly indicate the specific modes from the basis illustrated in Fig. A2.

See point below.

9. Related to comment #1. The single time step illustrated in the manuscript (figures 1 and A3) is better fitted with a selection of eigenmodes where the fundamental mode is the dominant. This is not the common situation, as shown in the associated movie. This needs to be discussed.

As shown in the temporal evolution analysis, now added to the paper, the temporal variation of the projection coefficients shows a non-stationary behaviour of the eigenmodes. However, some of them are more persistent than others and this can be also seen in the movies as well. This is now discussed.

10. In figure A1, indicate the selected sound speed value.

Done.

11. In the first paragraph from page 8, correct the color of the line. Orange instead of yellow.

Done.

12. In figure A2, the blue color means downward according to the chosen sign convention. I would recommend using blue color for upward since this is the general convention. Also, I understand that in figure 1 negative Doppler velocities (blue) correspond to upward.

We have removed that sentence because figure A2 just shows the modal basis. This is independent of the propagation direction. The direction of the propagation can be only determined after the modal projection and depends on the sign of the coefficients.

13. Indicate the time in both movies.

Done.

Reviewer #3 (Remarks to the Author):

Using IBIS observations of one of the largest sunspots of recent years, the authors have obtained a complex time series of its umbral oscillations over 3 hrs and sought to model them in terms of normal modes described by standard MHD theory. The results are broadly consistent, and on this basis the authors conclude that the standard theory gives an adequate and useful description of umbral waves.

There are several reasons for measuring and understanding umbral oscillations. One is to place seismic constraints on the important and mysterious subsurface structure, including dynamo action. This is extremely difficult, and the authors do not discuss it. Another is to provide boundary conditions for waves travelling from the umbra up to the chromosphere and beyond. Again, this is not discussed. Nevertheless, the presented results may find application in these areas.

We thank the referee for his/her suggestions which we have now mentioned in the paper as well as other possible applications, including the accurate modelling of the stellar noise in exoplanet detection, and asteroseismology in general.

I have a number of questions and comments.

1. On page 1, it is stated that sunspots are less subject to granular forcing due to their large sizes. However, this is more because of the suppression of convection by the strong magnetic fields.

We agree. Sentence removed.

2. Citing ref 18 on absorption of p-modes by sunspots, the authors say that p-mode absorption "appears as a natural explanation for the dominant (5-minute) power in sunspots". This is a misunderstanding of p-mode absorption. The absorption is actually a mode conversion process, at the layer where the sound and Alfvén speeds are equal, whereby the predominantly acoustic p-modes partially turn into magnetic waves that are either lost downward along the magnetic field lines into the solar interior, or upward as fast MHD waves into the chromosphere. In either case, energy is partially removed from those p-modes, and those that emerge from spots do so with reduced amplitude. This has nothing to do with whether or not spots exhibit that same 5-minute period peak as the quiet Sun. Instead, it is a matter that these quiet Sun p-modes are global normal modes of the Sun's surface layers, with distinct eigenfrequencies. They propagate around the Sun continuously, including inevitably through sunspots (where they are partially absorbed), so of course they will show wave power at these dominant frequencies.

We agree completely. The term "absorption" was used as it is sometimes used in the literature to refer to the reduction of power. But we agree with the referee that this is not correct. We have changed the term "absorption" with "conversion".

3. In Fig 2, is B the total magnetic field, or just its line-of-sight component, or just the vertical component? By the way, the eigenmode power peaks at 5.5 mHz and 9 mHz and $B=2500$ G are very interesting, and presumably tell us something.

It is the LoS component but we are very close to the disk centre. We have now specified that in the caption. Regarding the maximum power reached around 2500 G we have also noted that but we preferred not to comment on this as for larger values of B we may suffer from the S/N ratio due to the strong reduction of the photon flux. This may well be an effect of this and we opted to be on the safe side and not to over interpret the result.

4. On page 3: the vertical wavenumber k_z appears in m_0 in the wave equation. but little is said of it. That equation is actually only valid for tubes that are uniform along their lengths, the vertical direction in this instance. However, umbral magnetic field expands rapidly with height, so tube cross-section is certainly not uniform. More importantly though, the gravitational density scale height is only about 100 km, which is tiny in the context of a 30,000 km sunspot. So how is this equation relevant to umbrae. And if it is, what values of k_z have been used in the simulations and why?

The new approach to modelling applied here finds the solution for slow body modes using the observed umbral shape. The fact that there is such good agreement with the observed modes indicates that the umbral cross-sectional shape is not changing (or even expanding) very much over the integration height of the Fe 617.3 nm line (approx. 200 km). It is well known that the effect of gravitational stratification leads to the wavelength increasing with height. If there is no significant damping mechanism at play, then such stratification also leads to the amplitude of oscillations increasing with height. However, we are viewing the flux tube along its axis, over the integration height of 200 km so the v_z signal is an unavoidably integrated characteristic. Even for an expanding magnetic flux tube with gravitational stratification, if the cross-sectional shape and size is not changing very much over the integration height then the eigenmode spatial structure perpendicular to the flux tube axis will also not change very much. Here we are only trying to interpret the photospheric signal over a shallow integration height. If we were trying to connect both the photospheric and chromospheric layers of a sunspot umbra then the correct interpretation would rely much more on vertical stratification and the flux tube changing in shape and size changing with height, and there would be much larger integration heights with chromospheric lines to contend with. However, that is not the purpose of this current work.

Theoretically there is a continuum of possible eigenfrequencies for each slow body mode, and observationally due to the broadband nature of the driver, each individual mode has a broadband of frequencies associated with it. For the observed umbral shape the unique eigenvalue m_0 is numerically calculated for each mode. Physically, m_0 depends on the values of v_A , c_s , ω and k_z . From inversion estimates we get values for v_A and c_s . The analysed frequencies of interest range up to 10 mHz. Hence the unique value of k_z for each ω can be easily calculated. In the following plots we show phase speed vs k_z and frequency vs k_z in the frequency range of interest (up to 10 mHz). Since the slow body mode is very weakly dispersive, it can be seen in the frequency vs k_z plot, that all the modes when overplotted (in the frequency range up to 10 mHz) are practically sitting on a straight line.

5. The authors use a $v_z=0$ boundary condition at the umbra-penumbra boundary. It is by no means clear why this is appropriate. Indeed, it is not easy to see how it would be physically imposed by the Sun. Can the authors comment?

The observation shows that v_z oscillations decay very rapidly at the umbra-penumbra boundary. Theoretically, this is expected for a slow body mode, even if the environment is quiescent, which is certainly not the case for an actual umbra. We have used the $v_z=0$ boundary of the k-omega filtered data which is practically very close to the $B=1867$ G found in the literature. Had the observed modes been more consistent with surface modes then the $v_z=0$ boundary condition would not be valid since these modes have their maximum amplitude at the boundary. Some possible reasons we only detect slow body modes may be due to the nature of the driver, or perhaps there is an inherent “rigidity” of the umbra-penumbra boundary which prevents the excitation of surface modes, however at this stage this is purely speculation.

6. At the top of page 4, the authors refer to Fig 3. However, as far as I can see, there is no Fig 3.

Fixed

7. Page 4, paragraph "The remarkable agreement ..." does not really go to the question of which frequencies are prominent in the umbra. The 5.5 mHz and 9 mHz frequencies seen in Fig 2 are restricted to near the umbral centre. So their eigenfunctions should presumably be strongly peaked there. But no data on this has been presented. Fig A2 could be made much more useful by labelling each panel by its eigenfrequency, rather than the mode number.

As already said, the different modes have a similar spectrum.

8. According to the top left panel in Fig A3, the retained modes are 1, 13, 14, 16, 17, 18, 29, 34 and 36. However, this differs from those distinguished by people frames in Fig A2 (1, 12, 13, 15, 16, 17, 28, 33 and 35). By the way, purple is a poor choice of highlight colour, as it is difficult to see. I suggest red or dark green may be better.

We have tried to improve figure A2. The highlighted modes are now those that are the most persistent (i.e. and have more energy) over the entire observation.

REVIEWER COMMENTS

Reviewer #1 (Remarks to the Author):

The authors have provided clarification in the revised paper to address the questions in the original report.

Page 2 of methods, line 1 of text: Following Jess et al., the LOS Doppler velocity IS/WAS? filtered in the k-w space.

Reviewer #2 (Remarks to the Author):

I would like to thank the authors for the new version of the paper. The manuscript includes several improvements. Especially, the main strength of the work, which in my opinion is the construction of a physically constrained modal basis independent from the measured oscillations, is much clearly highlighted in the new version. Also, the paper now benefits from a richer discussion of the results. Unfortunately, I still do not think the results are convincing enough to grant publication.

My main concern is regarding the modeling of the power spectra. The authors detected a change in the power spectra from the quiet Sun/penumbra to the umbra in the B-w diagram. The appearance of new power peaks in the umbra is interpreted as the result of resonant modes, and an independent model is constructed based on this assumption. However, the model cannot explain those power peaks. In the new Fig. A4, the authors show the power spectra of the coefficients obtained from the individual fitting of the velocity at each time step to the constructed model basis. This is not sufficient to claim the relationship between the observed umbral power peaks and the modeled resonant modes. Their eigenfrequencies, and their relation with the observed power, need to be discussed. This link is fundamental to support the main result of the work, but it is not consistently addressed in the paper. This issue is argued in response to several of the referees' comments, but some of the points raised by the authors need clarification. For example, in the response to comment 4 from Reviewer #3 it is stated that "Theoretically there is a continuum of possible eigenfrequencies for each slow body mode". Do the authors mean that each of the eigenmodes illustrated in Fig. A2 is not associated with a single eigenfrequency, but with a continuum spectrum instead?

I also wanted to insist on the selection of the $v_z=0$ boundary condition. In response to comment 4 from Reviewer #2, the authors indicate "Had the observed modes been more consistent with surface modes then the $v_z=0$ boundary condition would not be valid". But Stangalini et al., ApJ, 869, 110 (2018) analyzed the same data set and detected surface modes in the umbra-penumbra boundary. Is this discrepancy due to the filtering in the k-w diagram? Please clarify.

Reviewer #3 (Remarks to the Author):

The manuscript is much improved, and I recommend publications. The results are certainly novel and of considerable interest.

Reviewer #1 (Remarks to the Author):

The authors have provided clarification in the revised paper to address the questions in the original report.

Page 2 of methods, line 1 of text: Following Jess et al., the LOS Doppler velocity IS/WAS? filtered in the k-w space.

Dear Referee,

Many thanks for the useful comments provided in the previous report. We are glad to hear that our clarifications have sufficiently addressed them all.

Regarding the minor comment at Page 2, we have fixed the text.

Thanks

Reviewer #2 (Remarks to the Author):

My main concern is regarding the modeling of the power spectra.

The authors detected a change in the power spectra from the quiet Sun/penumbra to the umbra in the B-w diagram.

The appearance of new power peaks in the umbra is interpreted as the result of resonant modes, and an independent model is constructed based on this assumption.

However, the model cannot explain those power peaks.

In the new Fig. A4, the authors show the power spectra of the coefficients obtained from the individual fitting of the velocity at each time step to the constructed model basis. This is not sufficient to claim the relationship between the observed umbral power peaks and the modeled resonant modes. Their eigenfrequencies, and their relation with the observed power, need to be discussed. This link is fundamental to support the main result of the work, but it is not consistently addressed in the paper.

This issue is argued in response to several of the referees' comments, but some of the points raised by the authors need clarification. For example, in the response to comment 4 from Reviewer #3 it is stated that "Theoretically there is a continuum of possible eigenfrequencies for each slow body mode". Do the authors mean that each of the eigenmodes illustrated in Fig. A2 is not associated with a single eigenfrequency, but with a continuum spectrum instead?

I also wanted to insist on the selection of the $v_{z=0}$ boundary condition. In response to comment 4 from Reviewer #2, the authors indicate "Had the observed modes been more consistent with surface modes then the $v_{z=0}$ boundary condition would not be valid". But Stangalini et al., ApJ, 869, 110 (2018) analyzed the same data set and detected surface modes in the umbra-penumbra boundary. Is this discrepancy due to the filtering in the k-w diagram? Please clarify.

Dear Referee,

Thanks for these additional comments.

Regarding the frequency of the eigenmodes, in our previous letter to the reviewers, we had included a calculation of the frequencies as a function of k_z , although we appreciate this was probably not detailed enough for the interested reader. In the revised version of the manuscript we have expanded this part, estimated the dispersion relation from the observational data and provided its comparison with the theoretical one. The corresponding figure and text are included in the "Methods" section.

In more detail, considering the particular umbral shape, the unique eigenvalue m_0 has been numerically calculated for each mode. Physically, these eigenvalues depend on the Alfvén and sound speeds, as well as on ω and k_z (already shown in the main text). The Alfvén and sound speeds were then modelled by employing state-of-the-art spectropolarimetric inversions, and used to compute the theoretical

dispersion relation. In this regard, the spectropolarimetric inversions themselves should be considered as part of our data-driven modelling approach, and this combination makes the most out of the current observational capabilities.

Theoretically, there exists a continuum of possible eigenfrequencies for each slow body mode (see e.g. Edwin & Roberts 1983) which fully agrees with spectral modelling performed in our study.

Due to the very weakly dispersive character of the slow body modes in the k_z range of interest, the obtained modes are practically sitting on a straight line (it can be seen in the frequency vs k_z plot, provided.).

This implies that each mode can assume a spectrum of frequencies. In this regard it is important to stress that **the B-Omega diagram of Fig. 2 does not only show a few more pronounced power features - corresponding to the modes with the larger amplitude - but also a series of additional (smaller) peaks** in between the most evident ones, which testifies to the presence of a wider range of excited frequencies simultaneously active as predicted by the model.

As stated in the updated version of the text, from the experimental point of view, the detection of all the frequencies (present in the system) depends on a number of factors, including the frequency resolution given by the total length of the data, the cadence and the intrinsic signal-to-noise ratio of the individual spectral features coupled with the instrumental sensitivity. For this reason, we should not expect to fully resolve all the spectral components of the physical system.

In order to further address the frequency modelling of the modes, **the dispersion relation of the observed modes has been estimated, by using simultaneous chromospheric data. The multi-height observations were used to infer the phase speed of the signals to be used in the dispersion relation. To the best of our knowledge it is the first time that such an analysis has been performed and, therefore, this adds to the other novel aspects of the observational results. We have found that the agreement between the observationally measured and modelled phase speed is remarkable. Furthermore, the theoretical and experimental dispersion relations agreed very well (within the uncertainties), and provide a full spectral characterization of the modes.**

Regarding the surface mode in Stangalini et al. 2018, we would like to stress that in this paper only downward propagating circular polarization fluctuations were detected and not Doppler velocity fluctuations. If any comparably large velocity fluctuation would have been present at the u-p boundary that would have been clearly visible in the B-Omega diagram as well, as a power feature localized in the proximity of the $B=1800$ G level and not extending to the center of the umbra (see, the right side of the diagram). But this is clearly not the case. In fact, the spectral features of the B-Omega go to zero in the proximity of the B value corresponding to the u-p boundary.

Reviewer #3 (Remarks to the Author):

The manuscript is much improved, and I recommend publications. The results are certainly novel and of considerable interest.

Dear referee,

We thank you very much for the valuable comments provided, which have certainly helped to improve the manuscript.

We are happy to hear that our work is recognized as novel and of considerable interest.

REVIEWERS' COMMENTS

Reviewer #2 (Remarks to the Author):

The authors have added a discussion about the dispersion relations computed from their modeling and the observations. The comparison shows a good agreement. The assumptions made for the calculations are reasonable, taking into account the inherent difficulty of the study. This discussion clarifies the issues raised in my last report.

Some minor comments:

-Please, revise the caption from Fig. A3. There are no top and bottom left panels.

-Figure 4 from Stangalini et al. (2018) shows a consistent phase lag between circular polarization and LoS velocity in the umbra-penumbra boundary, as pointed in my first report. I assume this is an indication of some velocity oscillations associated with the surface mode reported in that study. However, my doubts are probably more related to that work rather than this manuscript.

REVIEWERS' COMMENTS

Reviewer #2 (Remarks to the Author):

The authors have added a discussion about the dispersion relations computed from their modeling and the observations. The comparison shows a good agreement. The assumptions made for the calculations are reasonable, taking into account the inherent difficulty of the study. This discussion clarifies the issues raised in my last report.

Some minor comments:

-Please, revise the caption from Fig. A3. There are no top and bottom left panels.

-Figure 4 from Stangalini et al. (2018) shows a consistent phase lag between circular polarization and LoS velocity in the umbra-penumbra boundary, as pointed in my first report. I assume this is an indication of some velocity oscillations associated with the surface mode reported in that study. However, my doubts are probably more related to that work rather than this manuscript.

Dear referee,

we are glad to hear that the new analysis of the dispersion relations has clarified the issues raised in the previous report.

Here we resubmit our paper where caption of Fig A3 has been revised.

Thanks